# Blood Biomarkers for Alzheimer’s Disease in Down Syndrome

**DOI:** 10.3390/jcm10163639

**Published:** 2021-08-17

**Authors:** Laia Montoliu-Gaya, Andre Strydom, Kaj Blennow, Henrik Zetterberg, Nicholas James Ashton

**Affiliations:** 1Department of Psychiatry and Neurochemistry, Institute of Neuroscience & Physiology, The Sahlgrenska Academy at the University of Gothenburg, 431 41 Mölndal, Sweden; kaj.blennow@neuro.gu.se (K.B.); henrik.zetterberg@clinchem.gu.se (H.Z.); nicholas.ashton@gu.se (N.J.A.); 2Department of Forensic and Neurodevelopmental Sciences, Institute of Psychiatry, Psychology and Neuroscience, King’s College London, London WC2R 2LS, UK; andre.strydom@kcl.ac.uk; 3South London and Maudsley NHS Foundation Trust, London SE5 8AZ, UK; 4London Down Syndrome Consortium (LonDowns), London, UK; 5Clinical Neurochemistry Laboratory, Sahlgrenska University Hospital, 413 45 Mölndal, Sweden; 6Department of Neurodegenerative Disease, Queen Square Institute of Neurology, University College London, London WC1N 3BG, UK; 7UK Dementia Research Institute, University College London, London WC1E 6BT, UK; 8Hong Kong Center for Neurodegenerative Diseases, Hong Kong, China; 9Wallenberg Centre for Molecular and Translational Medicine, University of Gothenburg, Gothenburg, Sweden; 10Department of Old Age Psychiatry, Maurice Wohl Clinical Neuroscience Institute, King’s College London, London SE5 9RT, UK; 11NIHR Biomedical Research Centre for Mental Health, Biomedical Research Unit for Dementia at South London, Maudsley NHS Foundation, London SE5 8AF, UK

**Keywords:** biomarkers, Down syndrome, Alzheimer’s disease, blood, cerebrospinal fluid, positron emission tomography

## Abstract

Epidemiological evidence suggests that by the age of 40 years, all individuals with Down syndrome (DS) have Alzheimer’s disease (AD) neuropathology. Clinical diagnosis of dementia by cognitive assessment is complex in these patients due to the pre-existing and varying intellectual disability, which may mask subtle declines in cognitive functioning. Cerebrospinal fluid (CSF) and positron emission tomography (PET) biomarkers, although accurate, are expensive, invasive, and particularly challenging in such a vulnerable population. The advances in ultra-sensitive detection methods have highlighted blood biomarkers as a valuable and realistic tool for AD diagnosis. Studies with DS patients have proven the potential blood-based biomarkers for sporadic AD (amyloid-β, tau, phosphorylated tau, and neurofilament light chain) to be useful in this population. In addition, biomarkers related to other pathologies that could aggravate dementia progression—such as inflammatory dysregulation, energetic imbalance, or oxidative stress—have been explored. This review serves to provide a brief overview of the main findings from the limited neuroimaging and CSF studies, outline the current state of blood biomarkers to diagnose AD in patients with DS, discuss possible past limitations of the research, and suggest considerations for developing and validating blood-based biomarkers in the future.

## 1. Introduction

A child with Down syndrome (DS) born in the 1950s was expected to live less than 10 years, with congenital heart defects being the main cause of death [1]. Advances in medical care and improvements in the overall health have led to a dramatic increase in life expectancy of individuals with DS; for example, for such children born in 2010, the median life expectancy is estimated to be 65 years [2]. However, this longer lifespan is linked to an increased risk of developing dementia associated with Alzheimer’s disease (AD), with a prevalence of nearly 80% among those above 65 years old and becoming the leading cause of death in people with DS [3]. 

DS is caused by partial or complete triplication of chromosome 21, hence trisomy 21. The high risk for AD in DS is generally attributed to the triplication of the amyloid precursor protein (APP) gene encoded on chromosome 21. APP is a type I transmembraneous protein that is ubiquitously expressed with particularly high expression in neurons. It is cleaved by β- and γ-secretases into amyloid-β (Aβ) peptides, the main constituents of the amyloid plaques found in AD [4]. Overexpression of APP, and consequently overproduction of Aβ, results in an increased accumulation and deposition of Aβ in the brain. Amyloid plaques usually occur earlier in DS individuals compared with the general population, and deposits of Aβ in the cortex of DS individuals have been detected as early as at 12 years of age [5]. Although not all elderly individuals with DS receive a dementia diagnosis, nearly all individuals with full trisomy 21 aged 40 and older are found to have typical AD neuropathology [6], including extracellular amyloid plaques and intracellular neurofibrillary tau tangles, but also other features, such as cerebral amyloid angiopathy (CAA) [7]. The deposition of amyloid in leptomeningeal and cortical arteries in CAA can lead to microvascular events, which, given the frequency of CAA in DS, worsen the clinical presentation of dementia [8]. However, to date, a definite diagnosis of CAA can only be established with high certainty after a neuropathological examination, meaning that the prevalence of CAA is underestimated. 

Studies of rare case individuals with partial trisomy of chromosome 21 who have only two copies of the APP gene support the idea that AD in DS is driven by the extra copy of *APP*. In these individuals, postmortem neuropathological examinations revealed normal age-related changes but no evidence of AD neuropathology [9]. However, the triplication of other genes on chromosome 21 could also play a role in AD pathogenesis, as is suggested by findings of differing amyloid deposition depending on the extent of the triplication [10]. Genes that are present in chromosome 21 and are thought to promote AD pathology are involved in different molecular pathways: redox metabolism (*SOD1*), cholesterol metabolism (*ABCG1*), Aβ processing and clearance (*CSTB*, *BACE2* and *SYNJ1*), tau phosphorylation (*DYRK1A*), mitochondrial dysfunction (*RCAN*), and inflammatory responses (*S100B*, *IFNRs*) [11]. However, the extent to which triplications of these other genes affect AD pathophysiology in DS is not known, nor is whether the mechanisms are different from familial AD (fAD)-caused by mutations in *APP* or the genes encoding the enzymes that process APP (mainly presenilin 1 or 2) [12] or sporadic AD (sAD). Early clinicopathological and recent biomarker studies suggest that AD pathology as well as CSF and plasma biomarker changes in individuals with DS are qualitatively the same as in sAD and fAD [13,14,15]. Studies in the biomarker field are of notable relevance since early clinical diagnosis of AD by cognitive assessment is complex and challenging because patients with DS have a pre-existing and varying intellectual disability, which may mask subtle changes in cognitive functioning [16]. 

Cognitive screening tools traditionally used in medical settings to evaluate for AD dementia, such as the Mini-Mental Status Examination [17], are inappropriate for use within this population due to their premorbid cognitive impairments, which may affect scores. Measures to specifically evaluate cognitive decline among individuals with lowered intellectual functioning have been created, including the Dementia Scale for Mentally Retarded Persons (DMR) [18], Test of Severe Impairment (Albert and Cohen, 1992) and the National Task Group (NTG)-Early Detection Screen for Dementia [19,20]. As reviewed elsewhere in this Special Issue, specific screening tools to assess cognitive decline among individuals with DS have been further developed: Down Syndrome Mental Status Exam [21], the Cambridge Cognitive Examination for Older Adults with Down Syndrome (CAMCOG-DS) [22], the Cambridge Cognitive Examination for Mental Disorders of Elderly-Down Syndrome (CAMDEX-DS) [23], and the Cognitive Scale for Down Syndrome (CS-DS) [24], among others. However, clinical assessment and diagnosis remains key to diagnose AD in DS. 

The high prevalence of AD in DS makes this population an important target for frequent screening and monitoring for AD biomarkers. Cerebrospinal fluid (CSF) and positron emission tomography (PET), although accurate, are expensive, invasive, and particularly challenging in a vulnerable population such as people with DS. In contrast, collection of blood would be significantly easier for individuals with DS, making plasma or serum biomarkers the preferred option. In addition, blood examination is part of a routine clinical workup in DS, including tests for hypothyroidism and vitamin B12 deficiency. Biomarkers for AD could be included in routine workups to assist early diagnosis of dementia in DS and facilitate planning clinical care.

In this review, we summarize the main findings from neuroimaging and CSF studies in patients with DS and AD. We then discuss the use of AD-core biomarkers to diagnose AD pathology in DS patients and the recent progress made to find non-sAD-related biomarkers in blood. We conclude by examining possible limitations of studies in the past and by suggesting considerations to address in future investigations.

## 2. PET Biomarkers

Positron emission tomography (PET) studies with ligands for Aβ and tau are still few. Aβ accumulation is universal in those with DS over the ages of 40 years, as shown by postmortem studies [25]. Aβ ligand-binding is found in DS adults without clinical symptoms of AD but is highest in those with a dementia diagnosis [26,27]. Nonetheless, the relationship between the degree of Aβ binding and cognitive performance has been inconsistent [26,28,29], but it clearly increased with age [28,29,30,31], with great regional variability [32]. Based on Aβ PET thresholds, approximately 50% of individuals with DS between the ages of 40–50 years meet the criteria of AD. This increased to 90% in those >50 years old [33]. In contrast with sporadic AD, the progressive accumulation of Aβ in DS has a different temporal pattern. At the age of 40, in DS, Aβ accumulation begins in the striatum [26,34]—which is not seen for sAD but is consistent with fAD [35]. This early Aβ accumulation in the striatum has been suggested to be a consequence of Aβ overproduction [26] and a possible mechanism for early executive function and behavioral changes in DS. More consistent with sAD [36], Aβ positivity is seemingly needed for tau PET binding in DS. All DS individuals who are Aβ PET-negative are also tau PET-negative, and in a similar manner to Aβ PET, there is an exponential increase in tau after the age of 40 years [37]. Glucose hypometabolism via FDG PET has also been assessed in DS. In fact, cognitive dysfunction has been closely linked to changes in FDG PET signal [30,38], with DS individuals without dementia demonstrating the same hypometabolism pattern in the posterior cingulate as sAD [30]. In contrast, some DS patients demonstrate higher cerebral glucose metabolism in regions of grey matter atrophy. Therefore, it is proposed that increased glucose metabolism is a compensatory mechanism in the very early stages of dementia. Glucose hypometabolism is considered as a biomarker with a more linear change, with an initial decrease observed around 40 years old [13] (Figure 1).

## 3. CSF Biomarkers

The major advantage of cerebrospinal fluid (CSF) biomarkers is their direct contact with the brain, and therefore, CSF is an accurate reflection of central nervous system changes or developing neuropathology. However, the invasiveness of a lumbar puncture to obtain CSF restricts its collection in large cohorts of the DS population—which is represented by the few publications on this topic. The core CSF biomarkers for AD (Aβ42, p-tau, and t-tau)—which reflect Aβ deposition, neurofibrillary tangle pathology, and AD-related neurodegeneration, respectively [39]—are consistently changed in AD [40,41] and have therefore been in focus when studying AD in DS. In early childhood, Aβ levels tend to be elevated (due to increased APP expression and processing); later in life, the elevated levels begin to decrease in response to reduced clearance from the brain due to Aβ plaque development [13,42]. CSF Aβ42 concentration and Aβ42/40 ratio are lower in adult DS compared with age-matched controls [43], and the levels continue to decrease as amyloid pathology gets worse [44], which is paralleled by increased Aβ PET binding. In a longitudinal study on DS, CSF Aβ42/40 was found to change between 28 and 30 years of age, which is more than 20 years earlier than in sporadic AD and 10 years before Aβ PET changes [13]. DS individuals carrying the AD-associated apolipoprotein E (*APOE*) ε4 allele exhibited these changes earlier than non-carriers [45]. Other Aβ species (Aβx–38, Aβx–40, and Aβx–42) and soluble APP fragments (sAPPα and sAPPβ) showed higher concentrations in CSF from DS individuals than in CSF from healthy age- and sex-matched controls [46]. CSF tau (p-tau and t-tau) and neurofilament light (NfL) levels do not differ between DS and age-matched controls [44] but are increased in DS-AD [43] and correlate with age in DS [44,46]. In a multi-disorder study on CSF NfL [47], which included individuals with DS, CSF NfL levels were shown to be increased in the DS-AD group compared with non-demented DS. In comparison with sAD, there was no change. P-tau and NfL concentrations, however, increase after a change in CSF Aβ42/40, 10 years prior to an AD diagnosis—consistent with the pattern in fAD [13]. CSF NfL and the p-tau/Aβ42 ratio are reported to be the only biomarkers to significantly differentiate DS with mild cognitive impairment (DS-MCI) from control participants in a small pilot study [43].

Other CSF biomarkers reported to be changed in AD, YKL40 and SNAP25, have also been shown to be increased in DS-AD patients compared with healthy controls;while CSF α-synuclein, sTREM2, and Ng concentrations were not significantly changed [43] -but the study was likely lacking in sufficient statistical power. Recently, low CSF neuronal pentraxin-2 (NPTX2) concentration was reported in adults with DS. NPTX2, is a promising biofluid surrogate marker of inhibitory circuit dysfunction and cognitive decline in different dementias [48,49,50]. In DS, NPTX2 correlates with cortical atrophy and reduced glucose metabolism [51], but levels did not correlate with measures of cognitive decline. In contrast, an evaluation of multiple synaptic proteins in DS found that VAMP-2, which is predominantly found at glutamatergic synapses, was decreased in DS and correlated significantly with cognitive decline. However, in this study, CSF NfL and p-tau remained the best correlates of cognitive performance [52].

## 4. Blood Biomarkers

### 4.1. Blood Biomarkers Also Used in Sporadic AD

Biomarker results from CSF, PET, as well as from structural imaging, have allowed the pathophysiological definition of AD and the development of the “A/T/N” system, in which seven major AD biomarkers are divided into three categories based on the nature of the pathophysiology that each biomarker measures: amyloid pathology (A), phosphorylated tau (T), and neurodegeneration (N). Each biomarker category can be rated as positive or negative [39,53]. Currently available assays for sampling plasma ATN biomarkers appear to differentiate between AD patients, healthy controls, and non-AD dementias [54]. The ATN framework has also been proposed to be used to classify DS patients with dementia [55] (Table 1). 

#### 4.1.1. Amyloid: Plasma Aβ

So far, Aβ40 and Aβ42 peptides have been the most extensively investigated plasma biomarkers in DS. Studies comparing concentrations of circulatory plasma levels in individuals with DS with age-matched controls have repeatedly reported increased levels of Aβ40 and Aβ42 in DS [56,57,58,59,60].

Results regarding the Aβ42/40 ratio are less consistent. While several studies have reported this to be higher among individuals with DS as compared with healthy controls [61,62] and among those with DS and AD dementia as compared with those without dementia [61,62,63], a study by Schupf and colleagues identified that conversion to dementia over 4 years was linked to a decreased ratio of plasma Aβ42/40 [64]. Studies on the association between Aβ42 and age in DS have also shown contrasting results, with some studies finding no association [57,61,63,65,66,67], some finding a positive association [29,58,60], and some finding a negative association [56]. Additionally, the same was found for the correlation of Aβ levels and the stage of dementia progression. While some studies have suggested an association of cognitive decline with Aβ40 and Aβ42 levels [57,60,62,68,69], other studies have reported no differences in those with DS with and without dementia [61,62,63,64,66,67]. 

A meta-analysis of 10 different studies, including collectively 1482 individuals with DS, as well as 200 normal healthy controls, investigated the relationship between Aβ plasma concentrations and dementia in DS. Overall, individuals with DS were found to have increased plasma Aβ40 and Aβ42 levels compared with healthy controls. Individuals with DS who had a dementia diagnosis were found to have statistically higher plasma Aβ40 levels and lower Aβ42/40 ratios than non-demented DS. However, no significant association between plasma Aβ42 levels and dementia status was found [70].

The contradictory results reported in the past might be due to the use of varying and low-resolution methods. Similarly, until 2016 there was a wide variety of reports of plasma Aβ42/40 in sAD [41]. More recently, the use of ultra-sensitive technologies such as the single molecule array (Simoa), which can reliably measure ultra-low abundant proteins, has made the quantification of Aβ42/40 in blood more reliable [71]. In a longitudinal study with 100 individuals with DS and 100 age- and sex-matched controls, concentrations of Aβ42 were measured using Simoa technology. Across all ages, Aβ42 levels were shown to be increased in individuals with DS than in controls. Levels of Aβ42 decreased with age both in DS and controls, but this decrease was greater in DS and became prominent in the third decade of life [72]—this illustrates that these next-generation methods are beginning to mirror the Aβ findings in CSF.

Two other studies that used Simoa technology analyzed the diagnostic performance of Aβ40 and Aβ42 to discriminate different dementia stages in DS. In a first study with a high number of participants with DS (282 participants: 194 asymptomatic, 39 prodromal AD, 49 AD dementia), Fortea et al. showed that plasma Aβ40 concentrations were higher in the AD dementia group than in the asymptomatic group, but there were no other significant group differences for plasma Aβ42 (prodromal vs. asymptomatic, AD dementia vs. asymptomatic, and AD dementia vs. prodromal) or Aβ40 concentrations (prodromal vs. asymptomatic and AD dementia vs. prodromal) [15]. In a later longitudinal study with a slightly larger cohort (388 participants with DS: 257 asymptomatic, 48 prodromal AD, 83 AD dementia), the authors found that plasma Aβ40 and Aβ42 concentrations were significantly higher in all DS subgroups than in controls across the whole age span, but concentrations did not differ between diagnostic groups [13]. 

Lastly, another study used the Simoa technology to measure plasma Aβ40 and Aβ42 to understand relevant molecular differences between DS (*n* = 31), sAD without DS (*n* = 27), and controls (*n* = 27). Median concentrations of Aβ40 and Aβ42 were increased approximately 2-fold in individuals with DS compared with both those with sAD and controls. The Aβ42/Aβ40 ratio was higher in controls compared with DS or sAD, but there was no difference in the Aβ42/Aβ40 ratio between DS and sAD [73].

Lessons from plasma Aβ40 and Aβ42 sAD studies using Simoa technology do suggest that it is not the optimal method for determining cerebral amyloidosis, likely due to matrix effects. Recent data using full automated immunoassays for Aβ40 and Aβ42 or immunoprecipitation mass spectrometry (IP-MS) [74,75,76] for amyloid peptides have a superior correlation with either CSF or PET measures of Aβ. IP-MS technologies could also be more sensitive to Aβ accumulation at the prodromal phase of dementia [77] but, to date, have not been examined in DS. Another attribute of some IP-MS technologies is that some methods allow for measuring C-terminally truncated Aβ37 and Aβ38 peptides, which could be more informative in cases with CAA, where both Aβ40 and Aβ42 may be affected.

#### 4.1.2. Tau: Plasma Total and Phosphorylated Tau

Despite the relevance of tau in AD pathology, fewer studies have investigated the role of tau as a blood biomarker in DS dementia. This is likely due to the tau-encoding *MAPT* gene on chromosome 17 not being directly impacted by the triplication of chromosome 21, assays for total tau (t-tau) showing little promise in sAD, and assays of phosphorylated forms (p-tau181, p-tau217, p-tau231) having been only recently developed.

The plasma tau-related work that has been conducted in DS initially focused on the link between plasma t-tau in individuals with DS as compared with healthy controls. In this case, findings reported in the literature are consistent and indicate that plasma t-tau levels are increased in DS compared with healthy controls [13,15,68,78,79]. In addition, a positive correlation between age and plasma t-tau levels was found in two studies, but it was found only in the DS group, not in controls [68,78]. When plasma tau was measured using an assay (NT1) that detects forms of tau containing at least residues 6–198, levels were highest in children and fell with age in both the DS and control groups. In adults, the levels of NT1 tau overlapped in DS and controls until 30 years of age but tended to diverge afterwards. In individuals aged 50 years and older, NT1 levels were significantly increased in DS compared with controls [72].

However, when analyzing the possible role of plasma t-tau as a biomarker to diagnose AD and determine disease progression in DS patients, results point to a low diagnostic performance. Fortea and colleagues reported that plasma t-tau concentrations were significantly higher in DS-AD dementia compared with the DS-asymptomatic group and controls. There was weak evidence of higher t-tau concentrations in the prodromal AD group [15]. In their second study, the authors confirmed those results and observed increased plasma t-tau in people with DS and AD dementia compared with the asymptomatic group and controls, but there was overlap across the diagnostic groups [13]. Both of these studies on DS concluded that although CSF t-tau has a good diagnostic performance, it is poor in plasma. This is consistent with findings in sAD studies [80,81]. In a later study with 305 DS participants (225 cognitively stable (CS), 44 DS-MCI, and 36 DS-AD), t-tau produced an AUC of 74%, distinguishing DS-AD participants from CS, and an AUC of 56% for MCI-DS vs. cognitively stable (CS) [79]. 

Only two studies have compared t-tau levels between DS dementia and sAD. In the first one—which included 78 controls, 62 patients with AD, 35 with DS, and 16 with DS with degeneration—significant differences were detected in sAD vs. DS and sAD vs. DS dementia but not in DS vs. DS dementia. [68] Later, Startin et al. measured t-tau in adults with DS, adults with sAD, and controls using Simoa technology, without observing any significant difference among the groups [73]. 

As mentioned, results for sAD suggest plasma t-tau might not be an accurate biomarker for AD, while plasma phospho-tau species (p-tau), especially p-tau181 [82,83,84,85], p-tau217 [86], and p-tau231 [87], identify sAD and fAD pathophysiology with high accuracy and successfully discriminate AD from non-AD dementia cases. All these p-tau epitopes increase in the preclinical stage and correlate well with cerebral Aβ pathology; thus, they are of great interest in the development of amyloid in DS. In a publication with the same samples as for t-tau [78], Tatebe et al. reported that individuals with DS had significantly higher levels of p-tau181 compared with healthy controls [88]. Then, findings from blood-extracted neuronal exosomes revealed increased plasma phospho-tau (p-tau181 and p-tau396) in younger individuals with DS compared with age-matched controls, and the hyperphosphorylation persisted into late adulthood when DS concurred with AD dementia [89]. In a recent cross-sectional study by Lleó and co-workers, which examined 366 adults with DS (240 asymptomatic, 43 prodromal AD, 83 AD dementia) and 44 euploid controls, p-tau181 plasma demonstrated high diagnostic accuracy (AUC = 0.92) for differentiating controls and dementia. The accuracy was shown to be less when comparing controls with prodromal dementia (AUC = 0.80). This study concluded that plasma p-tau181 correlates with core fluid biomarkers of AD (CSF Aβ42/40, CSF t-tau, CSF NfL, and plasma NfL), as well as with atrophy in AD-related brain regions, including the temporal regions, angular and supramarginal gyri, and precuneus of both hemispheres (measured by MRI), and lower brain metabolism in temporoparietal regions (measured by FDG-PET) [90]. The latest research analyzing the association of the *APOE* genotype with AD biomarkers in patients with DS showed that *APOE* ε4 allele carriers showed earlier increases in plasma p-tau181, starting from the mid-40s and with confidence intervals not overlapping by age 50 years [45]. 

#### 4.1.3. Neurodegeneration: Plasma/Serum NfL

Neurofilament light chain (NfL) is one of the scaffolding cytoskeleton proteins of myelinated subcortical axons, and increased levels in CSF and plasma are an indicator of axonal injury due to neurodegeneration [91]. In contrast with Aβ and p-tau, altered levels of NfL do not specifically indicate the presence of AD pathology but are associated with brain atrophy and hypometabolism in AD [92,93]. This lack of disease specificity is less of a confounder in DS because most cases of dementia in DS patients are thought to be attributed to AD pathology. The development of ultra-sensitive assays to measure NfL in blood has facilitated the use of this biomarker for research purposes and its implementation in clinical routine [94].

In a first study in 100 adults with DS, blood NfL concentrations were increased with age, with a steep increase after 40 years. Moreover, NfL strongly associated dementia status in individuals with DS, and baseline levels were predictive of dementia diagnosis over time, even when adjusted for the strong effect of age [95]. A later study with 24 patients with DS and 24 controls showed significantly increased plasma NfL levels in the DS compared with the control group and a positive correlation between age and levels of plasma NfL in both groups. This age-dependent elevation was steeper in the DS compared with the control group. Moreover, elevated plasma NfL was associated with decreased adaptive behavior scores one year later, after age-adjustment [96]. Finally, in a longitudinal study performed by Mengel et al., it could be observed that for the first three decades of life, plasma NfL levels were relatively constant and were similar in DS and controls. In individuals over 30 years of age, NfL levels increased in both controls and DS, but the increase was considerably greater for people with DS [72].

Contrary to plasma Aβ and t-tau, the diagnostic performance of plasma NfL to distinguish AD stages in DS patients has shown promising results. In the clinical studies by Fortea et al., the diagnostic performance of plasma biomarkers was poor except for plasma NfL, which showed an AUC of 88% for the differentiation of the asymptomatic group vs. the prodromal AD group and 95% for the asymptomatic group vs. the AD dementia group. Plasma NfL concentrations were higher in all DS clinical groups than in controls. In addition, plasma NfL was increased more in both the prodromal AD and AD dementia groups than in the asymptomatic groups, but the comparison between prodromal AD and AD dementia groups was not significant. Among all the measured biomarkers, only NfL showed a strong correlation between plasma and CSF concentrations in participants with DS [15]. In their longitudinal study, which included PET measurements with FDG PET and amyloid tracers, they showed that the plasma NfL increase occurred almost 10 years before fibrillary Aβ deposition was detectable by PET and was one of the first biomarkers to change by age 28–30 years, more than 20 years before prodromal AD diagnosis [13]. In support of this, a recent analysis in >3000 individuals from multiple neurodegenerative disorders demonstrated that DS-AD had amongst the highest plasma NfL values in the study. Plasma NfL could also differentiate DS-AD from DS, with an AUC of 91%, 89% from Aβ-negative cognitively unimpaired controls and 100% from those with clinical depression. An age-dependent concentration cutoff of plasma NfL (19.37 pg/mL) could identify all DS dementia cases in this study [97]. However, in the study by Petersen et al., plasma NfL produced an AUC of 90% in distinguishing DS-AD participants from CS but only an AUC of 65% when distinguishing MCI-DS participants from CS. Interestingly, when plasma NfL was combined with t-tau, gender, and age, the model produced an AUC of 93% in distinguishing DS-AD participants from CS, which increased up to 87% when comparing MCI-DS vs. CS [79].

Plasma NfL levels’ association with PET biomarkers have been reported in a small cohort of 12 adults with DS, ages 30 to 60 years. Plasma NfL correlated with amyloid load as well as with markers of neurodegeneration (regional cerebral glucose metabolism—assessed with FDG PET—and hippocampal atrophy—assessed with volumetric MRI). Specifically, there were statistically significant relationships with plasma NfL in regions that are important to AD pathophysiology (i.e., precuneus and posterior cingulate gyrus). Increased levels of plasma NfL were also found to correlate with worse cognitive performance [98].

The ability of plasma NfL to reflect neurocognitive changes has been used to assess the validity of the DSM-5 criteria for neurocognitive disorder (NCD) in DS. NfL levels were increased in individuals diagnosed with NCD compared with those without NCD, despite no significant age differences between groups. This study points to the potential synergistic role of NfL and clinical criteria in improving the dementia diagnostic procedure in DS [99].

### 4.2. Blood Biomarkers for Non-ATN Processes in DS

Recent perspectives have challenged whether an increased genetic dosage of APP is sufficient to explain the highly increased AD susceptibility seen in individuals with DS [10]. Finding blood biomarkers that do not directly involve amyloid pathology (Table 2) may be of high value for (1) understanding broader mechanism of AD in DS and (2) providing with an earlier and/or more accurate diagnosis.

#### 4.2.1. Inflammatory Biomarkers

A potential involvement of the immune system in the development of AD is of particular interest in DS due to immune dysfunction being common in DS, with an increased vulnerability to some types of infections throughout life and higher rates of autoimmune disorders in DS (reviewed in [100]). In addition, the overexpression of immune genes found on chromosome 21 may point toward the role of systematic inflammation [101]. In adults and children with DS, higher blood (plasma or serum) concentrations of IL-10 [62,102], IL-6 [62,103,104], IL-1β [73], and TNF-α [62,103,105] have been reported compared with age-matched controls. A meta-analysis of 19 cytokine studies in adults and children with DS suggested that IL-1β, TNF-α, and IFN-γ (but not IL-6 or IL-10) concentrations are raised by trisomy 21 [106]. 

The alteration of plasma inflammatory biomarkers in patients with DS, even without AD symptoms, has been shown to remain altered in individuals with overt dementia. To further investigate the association between Aβ and inflammatory biomarkers with cognitive decline, Iulita et al. developed a linear regression model. They found that baseline plasma Aβ42 and the change in Aβ40, TNF-α, and IL-8 to be the best composite predictor of cognitive deterioration in their DS population. Therefore, the authors postulated that a combined measure of Aβ and inflammatory molecules was a strong predictor of prospective dementia [62].

Another inflammatory marker, the neutrophil gelatinase-associated lipocalin (NGAL), high blood levels of which are associated with risk factors for AD, was found to be increased in sera of elderly DS subjects compared with elderly non-DS controls [107]. Later, NGAL was quantified in 204 subjects with DS: DS with AD at baseline (*n* = 67), DS without AD (*n* = 53), and non-demented DS individuals that converted to AD (*n* = 84). NGAL was not associated with either diagnosed dementia or progression of dementia in DS. However, serum NGAL levels were associated with different plasma Aβ species according to the clinical symptoms of dementia in DS [108].

Although it has been demonstrated that the immune system is altered in both DS as well as in AD, how different the inflammatory profile in patients with DS-AD is when compared with sAD is not completely understood. Wilcock et al. demonstrated that DS-AD individuals have a different central nervous system (CNS) inflammatory phenotype compared with sAD cases. Even though DS-AD individuals exhibited strong elevations on M2b markers (e.g., CD86, CD64), these molecules were not significantly changed in individuals with sAD [109]. With the aim to study the differences in the plasma inflammatory profiles between DS-AD and sAD, Startin et al., apart from analyzing Aβ and t-tau as explained before, measured the levels of plasma IL1β, IL10, IL6, and TNFα in DS, sAD, and controls. Both AD and DS-AD groups exhibited high levels of IL-6, IL-10, and TNF-α, but IL1β concentration was far higher (10-fold) in those with DS compared with those with sAD and controls. Both the groups with DS and with sAD showed a moderate positive association between IL-10 and TNF-α concentrations, and the control group showed a moderate negative association between the Aβ42/Aβ40 ratio and IL-10 concentration. Within the cytokines investigated, there was a strong positive association between IL-1β and IL-10 and a moderate positive association between IL-6 and TNF-α. Finally, IL-1β concentration showed a moderate positive association with t-tau concentration and a moderate negative association with the Aβ42/t-tau ratio [73].

The relevance of the immune system in the development of AD in DS has also been demonstrated when developing plasma and serum proteomic profiles to distinguish MCI-DS and DS-AD from those who are CS. In a study with 305 participants (*n* = 225 CS; *n* = 44 MCI-DS; *n* = 36 DS-AD) enrolled in the Alzheimer’s Biomarker Consortium-Down Syndrome (ABC–DS) study, plasma and serum profiles scored an AUC ranging from 95% to 98% when distinguishing MCI-DS from those with CS and AUCs ranging from 93% to 95% in DS-AD vs. CS. Strikingly, biomarkers with a higher correlation (*R*^2^ > 0.9) included Eotaxin-3, IL-10, C-reactive protein, IL-18, serum amyloid A, and fatty acid binding protein 3; most of them are involved in the immune response and point to an alteration of the inflammatory profile in plasma in DS-AD [110].

#### 4.2.2. Cholinergic and Adrenergic Biomarkers

In AD and DS there is a progressive degeneration of basal forebrain cholinergic neurons [111,112], which depend on nerve growth factor (NGF) for their phenotypic maintenance [113]. Studies have demonstrated a compromise of the NGF metabolic pathway in AD [114] and DS brains [115], providing a mechanistic rationale to explain the degeneration of these neurons in such disorders. Plasma proNGF was measured by Western blot (WB) in 31 individuals with DS (with and without dementia) and 31 controls, and was shown to be increased in DS, even at AD-asymptomatic stages [62]. In a later study that quantified the levels of several NGF pathway proteins (proNGF, neuroserpin, tissue plasminogen activator (tPA), and metalloproteases) in samples from AD-asymptomatic DS (*n* = 14), prodromal AD (*n* = 10) DS, AD dementia (*n* = 12), and controls (*n* = 16), ProNGF and MMP-3 levels were elevated, while tPA was decreased in plasma in individuals with DS. The plasma levels of neuroserpin and MMP-9 were similar between all groups [116].

Levels of biogenic amines, such as increased noradrenaline (NA) and decreased serotonin (5-HT), have been shown to be altered in the CSF of AD patients [117]. Serum levels of the (nor)adrenergic metabolite 3-methoxy-4-hydroxyphenylglycol (MHPG), which diffuses freely over the blood–brain barrier, were significantly lower in demented (*n* = 51) and converted DS individuals (*n* = 50) compared with non-demented DS individuals (*n* = 50) and healthy controls (*n* = 22). Those with MHPG levels below the median had a more than 10-fold increased risk of developing dementia [118]. 

#### 4.2.3. Energy Metabolism and Oxidative Stress Biomarkers

Individuals with DS show intrinsic alterations of energy metabolism, with brain glucose hypometabolism noted in particular, which is similar to the findings of glucose hypometabolism as an early feature of AD [119]. A mass spectrometry (MS) study with 78 individuals with DS-AD and 68 non-demented DS patients showed a bioenergetically relevant alteration in peripheral blood plasma in DS-AD. Untargeted MS found significantly higher levels of lactic acid in the DS-AD group, which was confirmed by targeted MS. Additional targeted MS on central carbon metabolism revealed significantly increased levels of pyruvic and methyladipic acids, in addition to significantly lower levels of uridine in the DS-AD group [120].

Furthermore, deficits in mitochondrial function and oxidative stress play pivotal roles in DS and AD, and alterations in mitochondria occur systematically in both conditions [121]. Superoxide dismutases (SOD) are an important antioxidant defense in human cells exposed to oxygen, and the gene encoding the SOD1 isoform is located chromosome 21. A study with 32 DS patients showed that decline in memory performance over 4 years in adults with DS was positively correlated with SOD function measured at baseline [122].

#### 4.2.4. DNA Biomarkers

Some studies have suggested that AD causes an accelerated shortening of telomeres. Jenkins et al. showed in different publications that telomere length of T-lymphocytes from blood samples might be an accurate biomarker for MCI in adults with DS (MCI-DS). In total, they examined the sequential changes in telomere length in a total of 26 individuals with DS as they transitioned from preclinical AD to MCI-DS or dementia. Telomere shortening accompanied a change in the clinical status reflective of AD progression, but although the results are interesting, studies with larger cohorts are needed [123]. 

When investigating the relationship between DNA and aging, another factor that is commonly discussed is epigenetics. Haertle et al. studied the possible correlation of peripheral blood DNA methylation with cognitive impairment between individuals with DS and the increasing risk of developing AD. Patients with DS had different DNA methylation patterns compared with the general population: 2716 differentially methylated sites and regions discriminating DS and healthy individuals were identified. Among the significant changes, six sites were hypermethylated in both DS and AD patients vs. healthy controls. One of these was located in the ADAM10 promoter region, a gene that encodes the α-secretase, which is involved in the processing of APP [124]. 

**Table 2 jcm-10-03639-t002:** Blood biomarkers for non ATN-processes proposed for AD dementia in DS. Biomarkers are grouped according to the molecular pathway pathologically affected in DS. Colors for each group of biomarkers are related to Figure 2.

Affected Mechanism	Biomarker	Matrix	Technique	Platform	Results in DS Dementia	Refs
Inflammatory response	IFN-γ	Plasma	Mesoscale	Multi-Spot V-Plex Pro-inflammatory Panel (MesoScale Discovery)	DS ~ controls DS-AD > controls	[62]
	TNF-α				DS > controls DS-AD > controls	
	IL-6				DS > controls DS-AD > controls	
	IL-8				DS ~ controls DS-AD > controls	
	IL-10				DS > controls DS-AD > controls	
	NGAL	Serum	ELISA	ELISA kit (R&D systems) ELISA reader (Asys UVM 340 Biochrom)	Increased in DS. Association with species of Aβ depending on dementia progression	[108]
NGF metabolic pathway	proNGF	Plasma	WB	Semi-dry transfer (Bio-Rad)	DS > controls DS-AD > controls	[116]
	neuroserpin		WB	Semi-dry transfer (Bio-Rad)	Similar levels in all groups	
	tPA		ELISA	tPA ELISA kit (Abcam)	DS < controls DS-AD < controls	
	MMP-1		Mesoscale	Multi-Spot MMP 3-Plex Ultra-Sensitive kit SECTOR Imager 2400 (MesoScale Discovery)	DS > controls DS-AD > controls	
	MMP-3				DS > controls DS-AD > controls	
	MMP-9				Similar levels in all groups	
Biogenic amines	(nor)adrenergic (NA/A, MHPG)	Serum	RP-HPLC	Alexys^TM^ Dual Monoamines Analyzer	Lower levels MHPG in DS dementia vs. non-demented DS and heathy controls	[118]
	serotonergic (5-HT, 5-HIAA)					
	dopaminergic (DA, HVA, DOPAC)					
Carbon metabolism	Lactic acid	Plasma	Targeted LC-MS/MS	Triple Quadrupole MS (Xevo-TQ-S, Waters Corporation)	DS-AD > non-AD-DS	[120]
	Pyruvic acid				DS-AD > non-AD-DS	
	Methyladipic acid				DS-AD > non-AD-DS	
	Uridine				DS-AD < non-AD-DS	
Oxidative stress	SOD	Cytosolic and intracellular fractions from neutrophils	Spectro- photometry	Hitachi U-2010 spectrophotometer	SOD activity correlates to memory functioning in DS	[122]
DNA alterations	Telomere length	DNA extracted from T-lymphocytes	FISH	MetaSystems ISIS Image analyzer	Shortening of telomeres changes with AD progression in DS	[123]
	DNA mehylation	DNA extracted from whole blood	Sodium bisulfite conversion	Infinium MethylationEPIC BeadChips and Illumina iScan	2716 differentially methylated sites in DS, 9 related to dementia.	[124]

IL, interleukin; TNF, tumor necrosis factor; NGAL (neutrophil gelatinase-associated lipocalin); NGF, nerve growth factor; tPA, tissue plasminogen activator; MMP, matrix metalloproteinase; MHPG, 3-methoxy-4-hydroxyphenylglycol; SOD, superoxide dismutase; WB, Western blot; ELISA, enzyme-linked immunoassay; RP-HPLC, reversed phase-high-performance liquid chromatography; FISH, fluorescence in situ hybridization.

## 5. Past Limitations and Future Perspectives

Major advances have been made in the recent years in understanding DS-AD by investigating the evolution of imaging and fluid biomarkers. Indeed, several research groups have shown that there is a similar pattern of pathology between DS-AD, autosomal AD, and sAD [125]. However, the development of other pathologies in DS might promote AD progression and affect mechanisms that could alter the levels of other potential biomarkers. The establishment of reliable biomarkers is of particular interest in the population with DS due to the premorbid variability in intellectual disability severity posing an additional challenge to diagnose dementia clinically. Particularly, blood sampling is less invasive, easier, quicker, and cheaper to obtain than PET or CSF, often with limited feasibility to obtain in patients with DS. 

Despite the extensive research performed in the field, the high variability in results, especially in Aβ comparisons between DS and DS-dementia, has complicated reaching consensus on the possibility of using plasma biomarkers to diagnose AD in DS. One of the explanations for the contradictory results reported in the past might be the use of poor sensitive methods to measure low abundant biomarkers in blood. More studies using digital ultra-sensitive immunoassays or targeted IP-MS methods should be performed to clarify whether changes in Aβ peptides or ratios are associated with the development and progression of dementia in DS and to explore the possible use of plasma t-tau as marker for neurodegeneration. Furthermore, plasma GFAP has been recently reported to be a robust surrogate biomarker of amyloid pathology in sAD, independent of tau [126], and thus has potential to be a key biomarker in DS-AD. The emergence of p-tau181 as a biomarker for AD pathology is also showing usefulness in DS [90]. However, given that dementia in DS is highly likely to be due to AD, p-tau181 might not be superior to plasma NfL in this scenario—although more head-to-head studies including different p-tau epitopes (p-tau217 and p-tau231) are needed. Differences in study populations (including differences in age, dementia stage, or dementia duration) and sampling procedures might have also contributed to the discrepancies in the literature. A major advantage of plasma NfL is that it has high resistance to pre-analytical handling, meaning that the large variability in sample preparation often experienced in this population will have minimal effect [127,128]. Less is known about p-tau and Aβ, but they are more sensitive to freeze–thaw cycles than NfL [129]. 

Another possible limitation of some of the studies assessing AD blood biomarkers in DS has been the relatively small sample sizes. Cohorts required for these studies can be considerably smaller than those needed to achieve the same power for sAD owing to less variability in the phenotype for dementia, but clinical studies with relatively large samples are needed to test the diagnostic performance of the biomarkers. In addition, DS is much more common than autosomal dominant AD [130], and given the similarities between autosomal dominant AD and DS-AD, findings in biomarkers for AD-DS could potentially be exchangeable. 

Few studies have compared the performance of fluid biomarkers between DS-AD and autosomal AD and/or sAD [73,97]. Explorative proteomic studies would allow similarities and differences in their pathophysiology to be established, as well as to find biomarkers that could be more accurate for each type of dementia. In this case, CSF would be a preferable option since it better reflects changes in CNS than blood. If a specific protein proves to be of special interest, targeted assays could be developed in plasma with more chances of success. When analyzing proteomic profiles, it is worth mentioning the importance of longitudinal studies—assessing multiple time points starting prior to the onset of any cognitive decline—to help understand the evolution of the biomarkers. In a recent publication, Flores-Aguilar and colleagues showed that neuroinflammatory markers in DS brain increase during the first four decades of life, but interestingly, they decrease in late stages, possibly reflecting cell exhaustion and degeneration [131]. Determination of variations in the levels of the different biomarkers along the lifespan, under physiological and pathological conditions, would help to define clinical stages and pathophysiology of different AD types. 

Several efforts are being conducted to facilitate and standardize the research for DS-AD. The ABC-DS is a longitudinal study funded by the National Institute on Aging (NIA), with an aim to examine the progression of AD-related biomarkers (Aβ-, tau- and FDG-PET, MRI, CSF, plasma biomarkers, and neuropathology) as well as to examine cognitive functioning in over 400 adults with DS [132]. In Europe, the Horizon 21 Down Syndrome Consortium comprises various DS cohorts from the UK (the London Down Syndrome Consortium (LonDownS) and the Cambridge Dementia in Down’s Syndrome (DiDS) cohort), Netherlands (the Rotterdam Down syndrome study), Germany (AD21 study group, Munich), France (TriAL21 for Lejeune Institute, Paris), and Spain (the Down Alzheimer Barcelona Neuroimaging Initiative (DABNI). This large consortium is collecting longitudinal data on AD-related cognitive and clinical changes, along with standard AD biomarkers in participants with DS [13,15,125]. Finally, the Alzheimer’s Clinical Trial Consortium-Down Syndrome (ACTC-DS) study, funded by NIH, aims to serve as a platform for conducting clinical trials to treat and prevent AD dementia in individuals with DS [125]. 

The inexorable progression of AD pathology in the DS population makes it ideal for assessing the ability of biofluid proteins to reflect pathological changes occurring in the brain and thereby to support these alterations, potentially before any clinical decline is apparent. Biomarkers of preclinical AD-associated processes would be especially valuable since at this stage therapeutic intervention is most likely to be viable [133]. Knowing that most adults with DS will develop AD by their late 60s, this population would probably benefit from preventive therapies for AD, such as Aβ immunotherapies. The recent approval of aducanumab (Biogen) by the FDA opens up a new scenario for clinical trials to prevent AD in patients with DS [134].

## 6. Conclusions

Dementia in DS presents a major diagnostic and clinical management challenge because of the combination of learning disabilities, progressive cognitive and functional decline, and associated neuropsychiatric and behavioral symptoms. Reliable AD-biomarkers would greatly assist early diagnosis of dementia in DS. Blood-based biomarkers can potentially increase access via rapidly scalable, cost-effective, and minimally invasive methods that can complement and enhance other tools, such as cognitive assessment. In recent years, great advances have been achieved in the field of blood-based biomarkers for the diagnosis of sAD, and these have been explored for use in DS-AD. As it is the case in sAD, Aβ and total-tau, at least with the current available methods, have shown low diagnostic performance and overlap among groups. Additionally, plasma phospho-tau-181 has been proven to differentiate controls and dementia with high accuracy, but no better than NfL, unlike sAD. The high diagnostic performance of NfL, with good correlation with age and cognitive decline, in addition to its stability in pre-analytical handling and the fact that it can be measured by a robust assay used in clinical routine in many European countries, has pointed to plasma NfL as the best biomarker for diagnosis of AD in DS. However, more studies are needed to determine if other biomarkers, different than those used for sAD—for example, related to inflammatory, energetic or oxidative stress imbalances—could be earlier or more accurate biomarkers for dementia in DS (Figure 2).

## Figures and Tables

**Figure 1 jcm-10-03639-f001:**
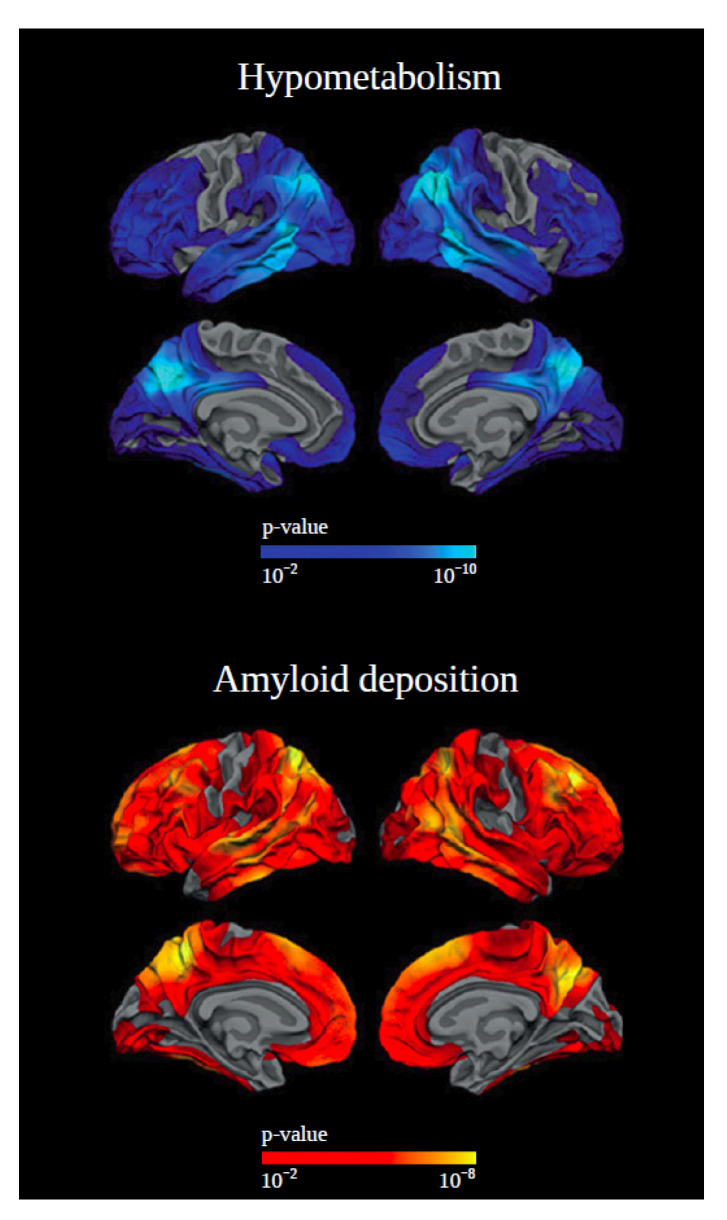
Example of positron emission tomography (PET) scans from individuals with Down syndrome (DS) with symptomatic Alzheimer’s disease (AD) compared with those with no clinical evidence of dementia. The upper panel represents a decrease in brain glucose metabolism of the temporoparietal, precuneus-posterior cingulate, and frontal brain regions in symptomatic DS. The lower panel shows the increased global cerebral Aβ deposition, with a relative sparing in sensory and motor areas in DS with symptomatic AD. Adapted with permission from [13].

**Figure 2 jcm-10-03639-f002:**
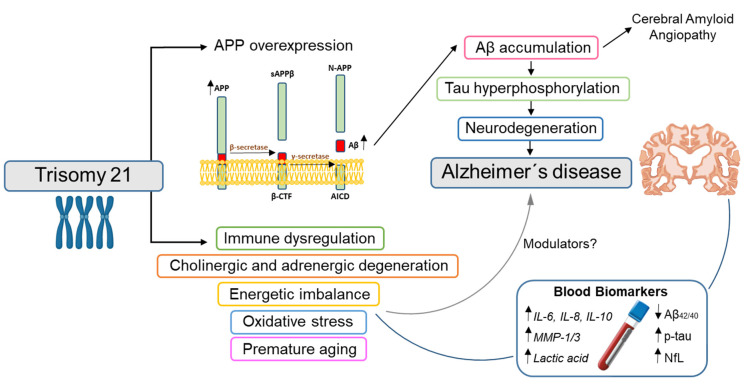
Schematic representation of the effects of trisomy 21 on vulnerability to Alzheimer’s disease and relation to potential blood-based biomarkers. Down syndrome (DS) is caused by partial or complete triplication of chromosome 21. The triplication of the amyloid precursor protein (*APP*) gene leads to overexpression of APP and consequently to overproduction of amyloid-β (Aβ) peptide. This results in an increased accumulation and deposition of Aβ (A) in the brain, that may promote tau hyperphosphorylation (T) and neurodegeneration (N). This ultimately translates in a higher risk to develop Alzheimer’s disease (AD). The triplication of other genes on chromosome 21, aside from *APP*, is associated with vulnerability to other pathologies, but findings suggest they could also play a role in AD. Colors of the box of affected mechanisms are linked to biomarkers in Table 2. To predict and diagnose AD in DS patients, blood biomarkersused in sAD (Aβ, p-tau and NfL) and biomarkers independent of A/T/N processes (italics) have been studied.

**Table 1 jcm-10-03639-t001:** AD core blood biomarkers studied in DS and DS-dementia. Each biomarker is categorized with (A), (T) or (N) depending on the pathology they reflect within the A/T/N system.

Biomarker	DS Compared with Healthy Controls	DS-AD Compared with Cognitively Stable DS	Diagnostic Application	Future Challenges
Aβ (A)	Increased levels of Aβ_40_ and Aβ_42_. Contradictory results for Aβ_42_/Aβ_40_ ratio and for association between Aβ_42_ and age.	Higher plasma Aβ_40_ and lower Aβ_42_/Aβ_40_ ratio in demented DS. No association between Aβ_42_ and dementia status.	Low diagnostic performance. Overlap between groups.	Quantification by sensitive IP-MS. Analysis of other Aβ species other than Aβ_40_ and Aβ_42_ (Aβ_37_, Aβ_38_, Aβ_41_)
p-tau181 (T)	Increased from early 30s. P-tau396 also found increased in neuronal exosomes.	Increased levels in both prodromal AD and AD dementia. Earlier increases in *APOE* ε4 allele carriers.	High diagnostic accuracy to differentiate AD dementia, low for prodromal AD.	Study of other tau phosphorylations: p-tau205, p-tau217, p-tau231.
T-tau (N)	Increased. Positive correlation between age and plasma t-tau in DS.	Significantly higher levels in DS-AD but weak increase in prodromal AD.	Low diagnostic performance. Overlap between groups.	New sensitive assays that target CNS specific tau.
NfL (N)	Increased. Positive correlation between age and plasma NfL.	Increased levels in both prodromal AD and AD dementia.	High diagnostic performance for both prodromal AD and AD dementia.	Use as diagnostic tool in clinical routine. Combination with cognitive assessment.

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
