# Peer review of "Blood Biomarkers for Alzheimer’s Disease in Down Syndrome"

_jcm, 2021, doi:10.3390/jcm10163639_

Round 1

Reviewer 1 Report

This review summarized the main findings from neuroimaging and CSF studies in patients with DS and AD. The authors conclude by examining possible limitations of studies in the past 113 and considerations to address in future investigations. The authors nicely reviewed recent leteratures published by Handen, et al.

I have some minor comments:

  1. Is there any recent information on imaging biomarkers for MRI in relation to AD core blood biomarkers studied in DS and DS-dementia?
  2. Tables 1 and 2. “Colors for each group of biomarkers are related to Figure 2.” There are no colors in these Tables.

Author Response

We would like to thank Reviewer 1 for the helpful suggestions. The answers for the comments are below:

  1. Is there any recent information on imaging biomarkers for MRI in relation to AD core blood biomarkers studied in DS and DS-dementia?

Yes, two publications have explored the relationship of brain atrophy measured with MRI with plasma p-tau181 and plasma NfL.

In a recent paper by Lleó and co-workers that describes the correlation between plasma p-tau181 and brain atrophy (measured by MRI) and hypometabolism (measured by FDG-PET) in temporoparietal regions. This was previously described in the paper in the blood-biomarker Tau section (lines 336-338), but for better clarification, that part has been re-written:  

This study concluded that plasma p-tau181 correlates with core fluid biomarkers of AD (CSF Aβ42/40, CSF t-tau, CSF NfL and plasma NfL), as well as with atrophy in AD-related brain regions including the temporal regions angular and supramarginal gyri and precuneus of both hemispheres (measured by MRI) and lower brain metabolism in temporoparietal regions (measured by FDG-PET)[90].

In a publication by  Rafii et al., the levels of plasma NfL were correlated to hippocampal atrophy measured by MRI. This information was also included in the previous version in the blood-biomarker NfLsection (lines 405-406), but has been rewritten for better understanding:

Plasma NfL correlated with amyloid load as well as with markers of neurodegeneration (regional cerebral glucose metabolism - assessed with FDG PET - and hippocampal atrophy – assessed with volumetric MRI-). Specifically, there were statistically significant relationships with plasma NfL in regions that are important to AD pathophysiology (i.e., precuneus and posterior cingulate gyrus). Increased levels of plasma NfL were also found to correlate with worse cognitive performance [98].

  1. Tables 1 and 2. “Colors for each group of biomarkers are related to Figure 2.” There are no colors in these Tables.

We apologize for the lack of colours in Tables 1 and 2, due to a formatting error. Hopefully, the final version will include colouring (or another association system if not possible). 

Reviewer 2 Report

The manuscript "Blood biomarkers for Alzheimer's disease in Down syndrome" by Montoliu-Gaya et al. comprehensively reviews the current knowledge related to biomarkers that can be measured in the blood to assess the propensity of individuals with Down syndrome to develop Alzheimer's disease. The choice of blood as a biomarkers source is well justified by relative non-invasiveness and inexpensiveness compared to, e.g., cerebrospinal fluid. The Authors show the pros and cons of long-studied (like Amyloid-beta, p-tau, NfL) and not so apparent biomarkers (e.g., inflammatory molecules or DNA methylation). Certainly, this work will be valuable for the scientific society researching Alzheimer's disease in humans with Down syndrome.

I suggest including the list of abbreviations in the paper, as this review is full of those, and such a listing would greatly help the reader. There are also several inconsistencies/spelling errors to be corrected (e.g., position/positron emission tomography; prodormal/prodromal; telemore/telomere). I could not see also coloring in the Tables as suggested by the captions. Nevertheless, those slight editing deficiencies do not reduce the high value of the manuscript.

Author Response

We would like to thank Reviewer 2 for the kind comments and helpful suggestions. We have included an abbreviation section in the beginning of the paper, and checked for inconsistences and spelling errors. We apologize for the lack of colours in Tables 1 and 2, due to a formatting error. Hopefully, the final version will include colouring (or another association system if not possible).